# Multimodal imaging and functional analysis of the chick NMDA retinal damage model

Tyler Heisler-Taylor[1,2], Richard Wan[1,3], Elizabeth G. Urbanski[1], Sumaya Hamadmad[1], Mohd Hussain Shah[1], Hailey Wilson[1], Julie Racine[4], Colleen M. Cebulla[1]*

**1** Department of Ophthalmology and Visual Sciences, Havener Eye Institute, The Ohio State University Wexner Medical Center, Columbus, Ohio, United States of America, **2** Department of Biomedical Engineering, The Ohio State University, Columbus, Ohio, United States of America, **3** College of Optometry, The Ohio State University, Columbus, Ohio, United States of America, **4** Nationwide Children's Hospital, Columbus, Ohio, United States of America

* colleen.cebulla@osumc.edu

## Abstract

### Objectives

The chick is rapidly becoming a standardized preclinical model in vision research to study mechanisms of ocular disease. We seek to comprehensively evaluate the N-methyl-D-aspartate (NMDA) model of excitotoxic retinal damage using multimodal imaging, functional, and histologic approaches in NMDA-damaged, vehicle-treated, and undamaged chicks.

### Methods

Chicks were either left undamaged in both eyes or were injected with NMDA in the left eye and saline (vehicle) in the right eye. TUNEL assay was performed on chicks to assess levels of retinal cell death one day post-injection of NMDA or saline and on age-matched untreated chicks. Spectral domain optical coherence tomography (SD-OCT) was performed weekly on chicks and age-matched controls day 1 (D1) up to D28 post-injection. Light adapted electroretinograms (ERG) were performed alongside SD-OCT measurements on post-injection chicks along with age-matched untreated controls.

### Results

Untreated and vehicle-treated eyes had no TUNEL positive cells while NMDA-treated eyes accumulated large numbers of TUNEL positive cells in the Inner Nuclear Layer (INL), but not other layers, at D1 post injection. Significant inner retina swelling or edema was found on SD-OCT imaging at D1 post-injection which resolved at subsequent timepoints. Both the INL and the inner plexiform layer significantly thinned by one-week post-injection and did not recover for the duration of the measurements. On ERG, NMDA-treated eyes had significantly reduced amplitudes of all parameters at D1 with all metrics improving over time. The b-wave, oscillatory potentials, and ON/OFF bipolar responses were the most affected with at least 70% reduction immediately after damage compared to the fellow eye control.

**Data Availability Statement:** All relevant data are within the manuscript and its Supporting information files.

**Funding:** The U.S. Army Medical Research Acquisition Activity, 820 Chandler Street, Fort Detrick MD 21702-5014 is the awarding and administering acquisition office. This work was supported by the Department of Defense, through the Vision Research Program under Award No. W81XWH1810805. Opinions, interpretations, conclusions and recommendations are those of the author and are not necessarily endorsed by the Department of Defense. In conducting research using animals, the investigators adhere to the laws of the United States and regulations of the Department of Agriculture. The authors acknowledge funding support from the Ohio Lions Eye Research Foundation (OLERF) and the Norbert Peiker Ohio Lions Eye Research Foundation Fellowship.

**Competing interests:** The authors have declared that no competing interests exist.

## Conclusion

This study establishes a normative baseline on the retinal health and gross functional ability as well as intraocular pressures of undamaged, vehicle-treated, and NMDA-damaged chicks to provide a standard for comparing therapeutic treatment studies in this important animal model.

## Introduction

Animal models are critical to understand the mechanisms of retinal disease and potential treatments. A damage model that has been commonly used in chicks and other species is the NMDA (N-methyl-D-aspartate) excitotoxic damage model [1–3], which mimics glutamate excitotoxic damage in the retina. Glutamate excitotoxicity occurs when a surplus of glutamate, a neurotransmitter, forms in the retina due to either excessive release or inadequate clearance by glutamine synthetase [4–6]. This excess glutamate activates NMDA receptors on inner retinal neurons which function as $Ca^{2+}$ pumps into the cells. The resulting increase in intracellular $Ca^{2+}$ acts as a second messenger and triggers a cascade that leads to eventual cell death [7–12]. This cell death can occur through both apoptotic and necrotic pathways depending on the glutamate or NMDA concentration [7]. This damage can be seen in diseases like glaucoma, retinal ischemia, vascular occlusion, diabetic retinopathy, sickle cell retinopathy, retinopathy of prematurity, and blast injury [7,13–16]. The NMDA retinal damage model simulates glutamate excitotoxicity by injecting NMDA into the vitreous cavity, which will activate the NMDA receptor, and cause $Ca^{2+}$ induced cell death [14].

In contemporary vision research, the domestic White Leghorn chick (*gallus gallus domesticus*) has become increasingly popular in serving as an ocular model that offers translational parallels to the human eye. The chick has several advantageous features and may currently be underutilized for ocular research [13]. Because the chick is an avian species that utilizes diurnal color vision [17,18], its eyes have a cone-rich retina [19–21] and a high-density region of photoreceptors called the area centralis [22–24], anatomically translatable to human's cone-rich retina and macula. The chick eye is much larger than the mouse eye, making it a more preferred model when administering intraocular injections. Chicks are also more cost efficient for high throughput testing [13].

Despite strides in the understanding of retinal cellular signaling during damage, there have been no multimodal studies comparing functional and structural changes in NMDA-damaged eyes vs controls. We seek to establish a baseline of biometric data in healthy and NMDA-damaged chicks, including key technology like the electroretinogram (ERG) to analyze the electrical function of retinal cells and spectral domain optical coherence tomography (SD-OCT) to view the retinal layers in live subjects. In our NMDA chick model, the greatest damage has been shown to be primarily localized to the inner nuclear layer (INL) [1]. We will also analyze intraocular pressure (IOP), weight, and cell death through the TUNEL assay so that future researchers may have a benchmark for future studies testing druggable targets to ameliorate excitotoxic damage.

## Materials and methods

### Animals

The use of animals in these experiments was in accordance with guidelines established by the National Institutes of Health and the principles of the ARVO Statement for Use of Animals in Ophthalmic and Vision Research. It was conducted under a protocol approved by The Ohio

State University Institutional Animal Care and Use Committee. Newly hatched wild type leghorn chickens (*gallus gallus domesticus*) were obtained from Meyer Hatchery (Polk, Ohio). Postnatal chicks were housed in a stainless-steel brooder at around 32˚C and kept on a 12-hour light, dark cycle. Water and Purina™ chick starter were provided *ad libitum*. Chicks were weighed prior to any treatments or sacrifice timepoints. Slit lamp observations and fundus imaging with an operating microscope were taken prior to any measurements. These studies utilized 4 chicks for TUNEL analysis, 20 chicks for the initial round of ERG/OCT/etc testing, and 14 chicks for the supplemental ERG/OCT/etc testing making a total of 38 chicks.

## Chick anesthesia

The chicks were kept warm to maintain an internal temperature of 40.3–40.8˚C via a heating pad and blanket (a double folded cut of surgical cloth) during surgical and clinical procedures and under a heat lamp at all other times.

The chicks underwent general anesthesia via a Kent Scientific VetFlo Calibrated Vaporizer. The anesthesia exposure duration was recorded to mitigate overexposure and potential harm to the animals.

## Chick injection

5% betadine, an antiseptic topical solution, was prophylactically administered on the ocular surface of each eye, and then the superior eyelid and periocular region were swabbed with betadine solution via a cotton-tipped applicator. The betadine was left to dry for at least 30 seconds before inserting a 25μL Hamilton syringe or a 29Ga Insulin Syringe through the superior palpebra penetrating the globe into the posterior segment. 20μL of NMDA (25mM) was slowly injected intravitreally into the left eye (OS) and the syringe was slowly withdrawn from the globe approximately 5 seconds after the injection was complete [25,26]. The eye was then rinsed with sterile saline to wash out any residual betadine. The same procedure was performed on the fellow chick eye (right eye, OD) with sterile saline, the vehicle control. Chicks were injected and sacrificed at three timepoints. Chicks were injected at post-hatch day 7 (P7) and sacrificed one day after injection (D1) for cell death (TUNEL) analysis, sacrificed two weeks after injection (P21 or D14), and sacrificed 4 weeks after injection (P35 or D28).

## Tonometry

The iCare Tonovet Plus was used to measure intraocular pressure on chicks before and after each injection and weekly before any other operations. Post-injection measurements were taken 3–10 minutes afterwards in order for the pressure to normalize. The iCare Tonovet Plus was used on the lapine or rabbit setting and was found to produce similar readings to those reported in the literature [27–31]. The iCare Tonovet Plus was operated according to the manufacturer's instructions with both eyes of each chick being measured consecutively over an average of five readings.

## Spectral domain optical coherence tomography

SD-OCT was performed on chicks on both eyes weekly after the initial injection of NMDA using an Envisu R2200 SDOIS, a rabbit bore lens, and the InVivoVue software. Prior to beginning, chicks were anesthetized via isoflurane following the protocol previously stated. A chick was then placed on an imaging mount and the nose cone of the isoflurane line was secured at the rat bite bar to sustain controlled anesthesia throughout the process. To maintain a continual clear view, the chick eye was secured open with a small, lid speculum (2mm barraquer wire

eye speculum) and hydrated with Systane Ultra drops, as needed. The chick position was adjusted until the pectin, located inferiorly in the retina, was positioned in the central inferior view on the imaging program (S1 Fig).

On each eye, three scans were acquired: a 12mm 1000x1x60 frames (1000 A-scans per B-scan, 1 B-scan, 60 B-scans per frame) linear B-scan for high quality images, a 12x12mm 1000x6x25 frames radial scan for data collection, and a 12x12mm 400x400x4 frames volumetric scan for volume intensity projection (VIP) views. The captured scans were then averaged in InVivoView and analyzed in InVivoVue Diver. In the Diver software, 8 measurements points were selected (S1B Fig) approximately 1-2mm from the pecten allowing the measurements of various but consistent retinal thicknesses (S1C Fig).

## Electroretinography

All ERGs were performed under the guidance of a trained visual electrophysiologist. Light adapted ERGs were first recorded with the use of the UTAS 3000 system (LCK Technologies Ltd.). Light adapted ERG were also recorded with the Celeris (Diagnosys, LLC) system on an independent set of chicks. For the duration of the testing, chicks were kept anesthetized (isoflurane; as mentioned above) with rectal temperature monitored and maintained between 40.3–40.8˚C.

For the UTAS system, a ground wire needle (Grass E2 subdermal electrode) was subcutaneously inserted under the wing and the reference electrode needle (Grass E2 subdermal electrode) at the posterior end of the head. A drop of 0.5% tetracaine was administered topically to provide corneal anesthesia. A DTL Plus Electrode (Diagnosys, LLC) was positioned on the chick such that the electrode fiber rested over the pupil of the eye. Genteal Severe Gel or Refresh Celluvisc was topically administered to keep cornea moist during testing. Each chick was laid on their side and the chick's eye was positioned centrally inside the ERG ganzfeld. Right eye was tested first followed by the left eye. However, with the Celeris system, no subdermal electrodes were required. Genteal Severe Gel was topically administered to the Celeris bright flash stimulators (5mm apparatuses) to facilitate electrical contact and keep the cornea moist. Stimulator were placed on each eye as per the manufacturer's recommendations and both eyes were tested in an alternating manner.

Light adapted electroretinograms were recorded every 7 days from post hatch day 7 to 35. Electroretinograms (bandwidth (UTAS): 0 to 500 Hz, bandwidth (Celeris): 0.125 to 300 Hz) and oscillatory potentials (bandwidth (UTAS): 75 to 500 Hz, bandwidth (Celeris): 75 to 300 Hz) were recorded simultaneously. The light adapted ERGs and OPs (background: 30 cd/m$^2$) were evoked to flashes of white light (flash duration less than 4 ms) spanning over a 2 log unit range with a minimal intensity of 0.062 (Celeris: 0.05) and maximal intensity of 25 cd.sec/m$^2$. Averages of 20 responses were taken at each step. An interstimulus interval (ISI) of 1 sec was used throughout the testing period.

Amplitude of the a-wave was measured from the baseline to the first negative trough while that of the b-wave was measured from the a-wave trough to the most positive peak of the ERG. The amplitude of all OPs was also summated to yield the OPSum value. Peak times were measured from the flash onset to the peak of each component. Light adapted flicker ERGs were also recorded. Background was set at 30 cd/m$^2$ with a flash intensity of 4.4 cd.sec/m$^2$ (Celeris: 5 cd.sec/m$^2$) and a frequency of 15, 20 and 25 Hz. Amplitude was measured from the preceding through to each positive peak. Long flash stimulation was also recorded. Background intensity was set at 30 cd/m$^2$ with flash intensity of 4.4 cd.sec/m$^2$ (Celeris: 5 and 10 cd.sec/m$^2$) with ISI of 5 seconds and a flash duration of 150 ms. Amplitude of the ON response a-wave was measured from the baseline to the first negative trough while that of the ON response b-

wave was measured from the a-wave trough to the most positive peak of the ERG. Amplitude of the OFF responses was measured from the flash offset to the most positive peak after the flash offset.

## TUNEL

TUNEL assay was performed according to the manufacturer's instructions (TMR red kit, cat#: 12156792910, Roche) to detect cell death in the retina INL. The assay stained at 568nm (red) while a nuclear counterstain was applied with DAPI at 461nm (blue). Images were taken with a Leica DM5000B fluorescent microscope and Leica DC500 digital camera at 200x magnification with the Leica Application Suite 4.8.0 software. Exposure settings were adjusted to minimize oversaturation. Cells were counted utilizing the MCT method [32].

## Statistical analysis

Data were analyzed in a masked fashion and calculated in Microsoft Excel and JMP. ERG data were processed and analyzed in Graphpad PRISM before entry into JMP for statistical analysis. The Kruskal-Wallis non-parametric test was performed to evaluate differences between experimental groups and the Steel-Dwass non-parametric post-hoc testing for multiple comparisons. Error bars are listed with standard error of the mean (SEM) unless stated otherwise and an $\alpha = 0.05$.

# Results

## Clinical observations

Slit lamp observations of the external ocular surface and anterior segment as well as fundus imaging of the posterior segment and retinal surface showed no media opacity on undamaged and vehicle eyes. Most eyes displayed some degree of punctate epithelial erosions, likely due to the process of repeated measurements. In NMDA-damaged eyes, while it was difficult to find the pecten, it was clear and sharp indicating a lack of media opacity, however the retina itself had changed from pink with visible scleral veins prior to any injections (Fig 1A) to a foggy white coloration (Fig 1B). NMDA damage was not found to impact chick weight gain (S2 Fig).

## Intraocular pressure

IOP was measured before and after injections and weekly up until sacrifice on undamaged, vehicle (saline) injected, and NMDA-damaged eyes (n = 10/group D1-D14, n = 5/group D21-D28, Fig 2). While NMDA trended lower than controls at each of the timepoints, it was only significantly lower than undamaged eyes at the P7/D1 timepoint (Undamaged: 15.20 ±0.89mmHg vs. NMDA: 13.74±0.73mmHg, p = 0.0075). Additionally, there were significant increases over time as the chick aged in all three conditions (NMDA, vehicle, and undamaged). IOP variations were within the clinically acceptable range of only few mmHg in difference and could have been due to a multitude of factors including normal diurnal changes to the IOP, difference in researcher measurement technique, and tonometer variation. When comparing the IOP measurements of NMDA and vehicle (saline) treated eyes from pre- and post-injection measurements, we saw no significant differences whether immediately after the injection or a day later.

## TUNEL

Select representative images were chosen to display the difference between untreated (n = 4), vehicle-treated (n = 4), and NMDA-treated TUNEL fluorescent captures (n = 4, Fig 3).

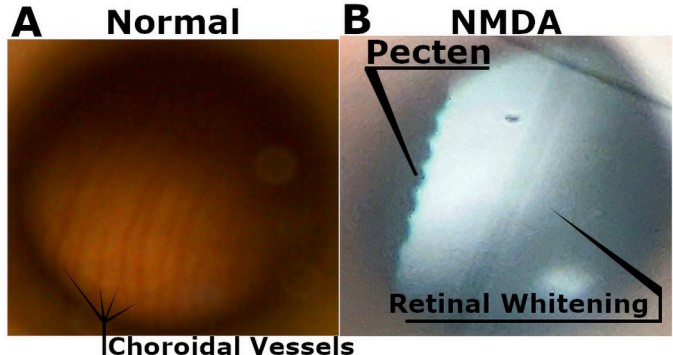

**Fig 1. Representative fundus imaging.** Choroidal vessels are visible in healthy chick eyes (A). While in NMDA-damaged chick eyes they can no longer be observed, instead we see the retina assume a homogenous white color potentially due to a lack of perfusion (B). The sharp and clear presence of the pecten rules out media opacity in NMDA-damaged eyes.

Untreated and vehicle eyes were observed to contain no TUNEL positive cells indicating the lack of cells undergoing either apoptosis or necrosis. NMDA-damaged eyes however exhibited significant accumulation of TUNEL positive cells (3624.94 ± 449.51 vs. 0.00 ± 0.00 TUNEL positive cells/mm$^2$, p = 0.0211 vs untreated).

## SD-OCT

Chicks treated with NMDA (OS) and vehicle (saline, OD) as well as age-matched undamaged controls were imaged with SD-OCT at D1, D7, D14, D21, and D28 with retinal thickness calculated from the resulting B-scans (Fig 4). Retinal thickness was calculated for the retinal nerve fiber layer (RNFL), inner plexiform layer (IPL), a combination of the RNFL and IPL,

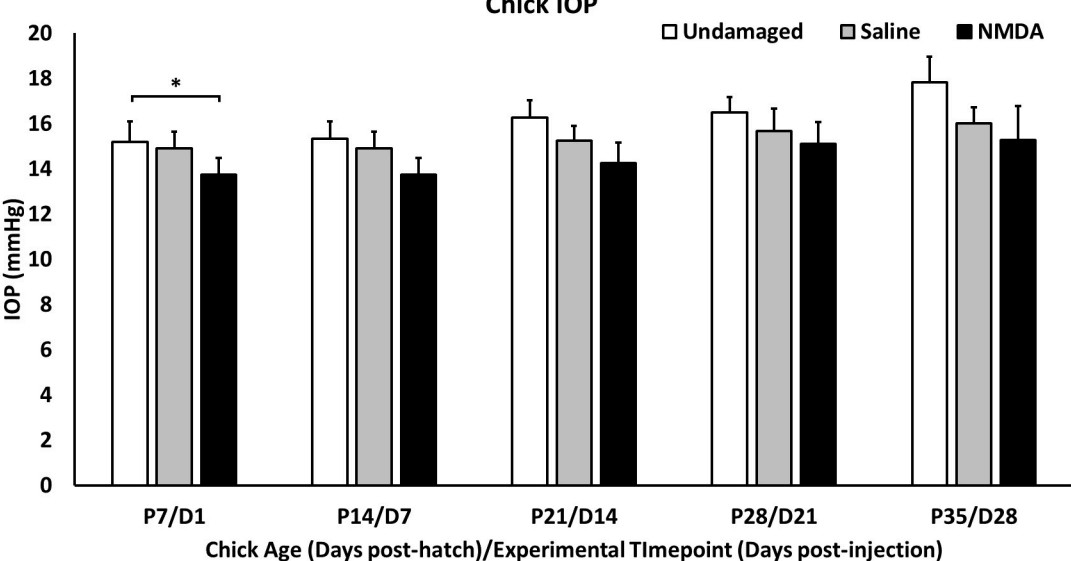

**Fig 2. IOP measurements.** Chick IOP measurements on undamaged chicks and NMDA-damaged chicks (OD: Saline, OS: NMDA) taken weekly starting at post-hatch day 7 or post-injection day 1 till post-hatch day 35 or post-injection day 28.

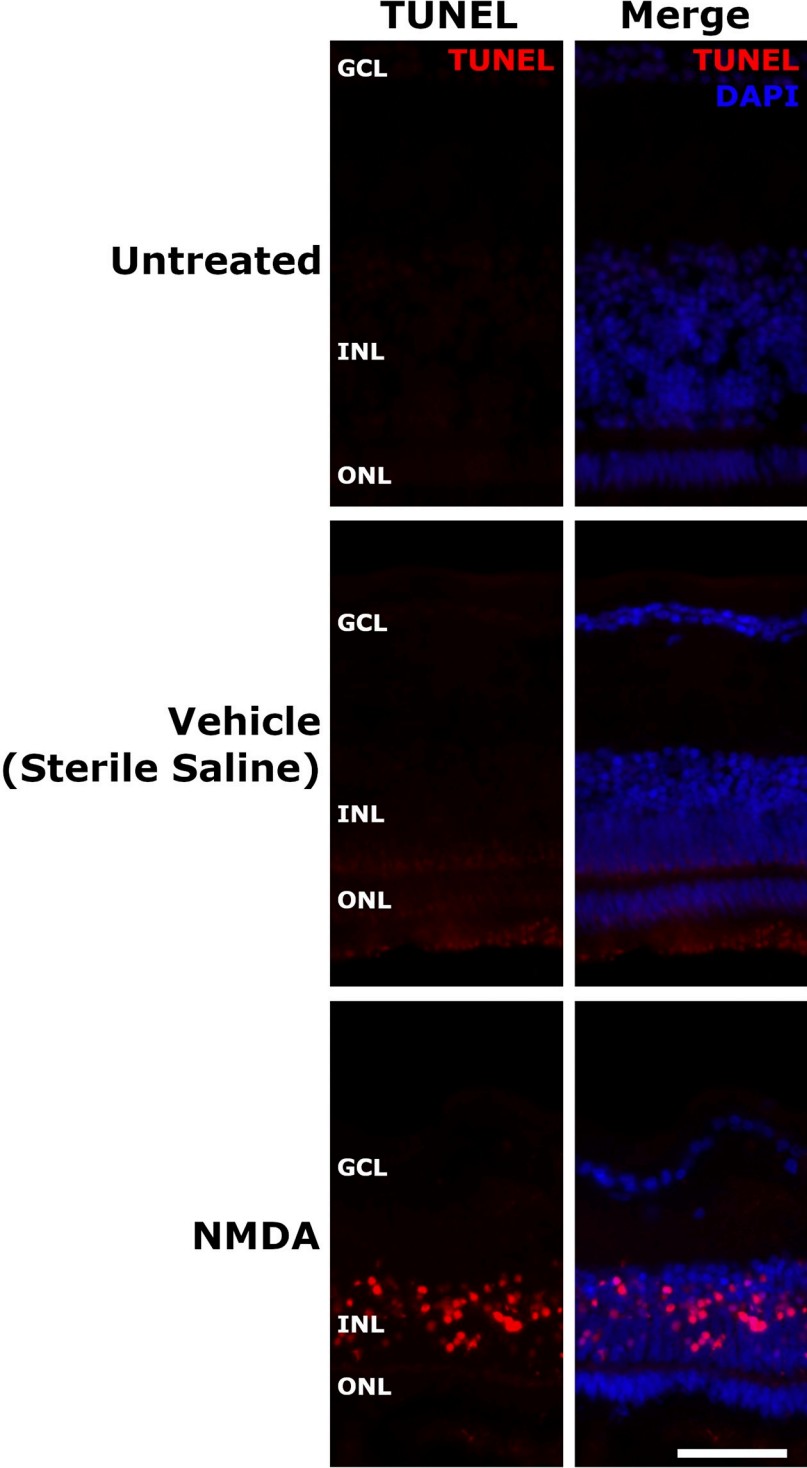

**Fig 3. NMDA induced TUNEL response in chick retina.** Untreated, vehicle (sterile saline), and NMDA injected chicks at D1 post injection. Both untreated and vehicle-injected chicks displayed no TUNEL-positive cells. NMDA-treated chicks saw a large number of TUNEL-positive cells in the INL, but not other retinal layers. The scale bar denotes 50μm. Abbreviation: GCL—ganglion cell layer, INL—inner nuclear layer, ONL—outer nuclear layer.

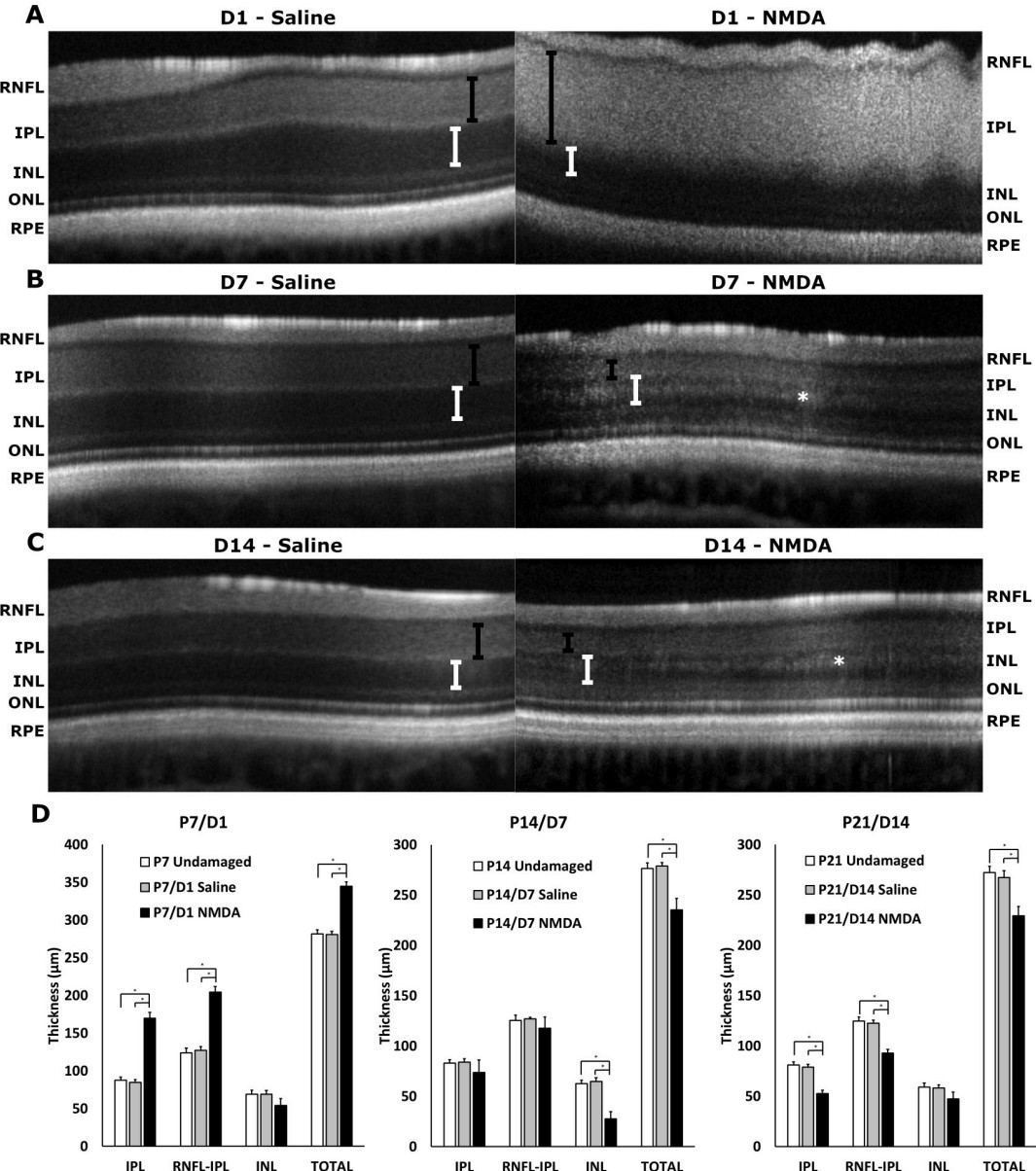

**Fig 4. SD-OCT analysis of NMDA-treated and corresponding control eyes.** Representative b-scans of D1 vehicle (saline, left) and D1 NMDA (right) showing edema in IPL (A), D7 vehicle (saline, left) and D7 NMDA (right) (B), and D14 vehicle (saline, left) and D14 NMDA (right) (C). (D) Analysis of retinal thicknesses of the IPL, RNFL-IPL combined, INL, and total thickness between NMDA and saline (vehicle) treated eyes at P7/D1 to P21/D14 post injection and their respective age-matched untreated controls. Black brackets show IPL thickness, white brackets show INL thickness, asterisk denotes hyperreflective layer. Error bars are shown as standard deviation. Abbreviations: RNFL—retinal nerve fiber layer, IPL—inner plexiform layer, INL—inner nuclear layer, ONL—outer nuclear layer, RPE—retinal pigment epithelium.

INL, outer plexiform layer (OPL), outer nuclear layer (ONL), external limiting membrane (ELM), photoreceptors (PR), retinal pigment epithelium (RPE), and total thickness. Significant differences between NMDA-damaged eyes and controls were found in the IPL, INL, and total thickness (Fig 4D). Other layers were found to be unchanged (S2D Fig). Average thickness

values were used to compose a model to visually display the morphologic changes in the retina over time (Fig 6A).

The significant differences observed at D1 post-injection (Fig 4A) were most evident in the IPL with the presence of an inner retinal edema in the NMDA-treated eyes (Saline: p = 0.0151, Untreated: p = 0.0151, Fig 4D). Due to this edema, the RNFL-IPL combination and total retinal thicknesses of NMDA-treated eyes were significantly increased over controls. The INL appeared to begin thinning at D1 however the different was only borderline significant (Saline: p = 0.0545, Untreated: p = 0.0545, Fig 4D). One-week post-injection (D7) the retinal swelling had resolved and the effected layers became significantly thinner (Fig 4B). Most affected were the IPL becoming non-significant (Saline: p = 0.5863, Untreated: p = 0.6977, Fig 4D), the INL becoming significantly lower than controls (Saline: p<0.0151, Untreated: p<0.0151, Fig 4D), and the Total retinal thickness going from significantly higher to significantly lower than controls (Saline: p<0.0151, Untreated: p<0.0151, Fig 4D). The retinal thinning continued at D14 as the IPL and RNFL-IPL displayed the greatest reduction becoming significantly lower than controls (Saline: p = 0.0151, Untreated: p = 0.0151, Fig 4C). This thinning in the IPL counteracted the apparent reduction in thinning observed in the INL resulting in the total thickness remaining roughly the same as the previous timepoint, potentially a result of a new hyperreflective region within the INL (Fig 4B and 4C). The pattern observed at D14 continued during the subsequent timepoints at D21 and D28 (Fig 6A).

## Electroretinogram

The a-wave serves as a metric for the function of cone photoreceptor and cone OFF-bipolar cells [33]. NMDA treatment was found to have the most significantly reduced a-wave amplitudes at D1 (38% reduction at 5cd.sec/m$^2$) with recovery at later timepoints (10% reduction at D28 at 5cd.sec/m$^2$) as NMDA appeared to recover towards controls (Fig 5 –top row and Fig 6B). The b-wave, which serves as a metric for the functionality of bipolar cells [33], was expected to show the most impairment in the NMDA damage model due to excitotoxic damage preferentially affecting bipolar and amacrine cells (Fig 5 –second row and Fig 6B) [25]. At D1, the NMDA-damaged eyes were significantly different from both controls at almost every flash intensity (75% reduction at 5cd.sec/m$^2$). From D7 through to D21, the damage profile remained similar with a 44% reduction at 5cd.sec/m$^2$ across these timepoints. By the final timepoint, the retinal function had restored up to a 30% reduction at 5cd.sec/m$^2$.

Oscillatory potentials, which summarizes the functionality of amacrine cells [33], were summed and compared similar to the a- and b-wave (Fig 5 –third row and Fig 6B). Akin to the b-wave, there was a large reduction of amplitudes at D1 (63% reduction at 5cd.sec/m$^2$) that did not show recovery like the a-wave (38% reduction at D28 at 5cd.sec/m$^2$). The flicker response, which encapsulates the function of the cone pathway [33], was not found to display significant differences between 15, 20, and 25Hz. The 20Hz data was chosen to show the differences between NMDA, vehicle, and undamaged eyes (Fig 5 –fourth row and Fig 6B). At most timepoints, NMDA was significantly lower than both controls (D1: 62% reduction, D28: 40% reduction).

The long flash ON/OFF response test allows separate measures for the functionality of the ON and OFF cone bipolar cells. The ON-bipolar response was found to show significant differences between NMDA damage and both controls from D1 (89% reduction) to D14 (68% reduction). At D28, the amplitude had recovered to only a 32% non-significant reduction. The OFF-bipolar response was only significant between NMDA and both controls at D1 (71% reduction) and D7 (42% reduction). By D28, the amplitude reduction had fallen to 15% (Fig 5 –bottom row and Fig 6B). This would indicate that ON bipolar cells are potentially more

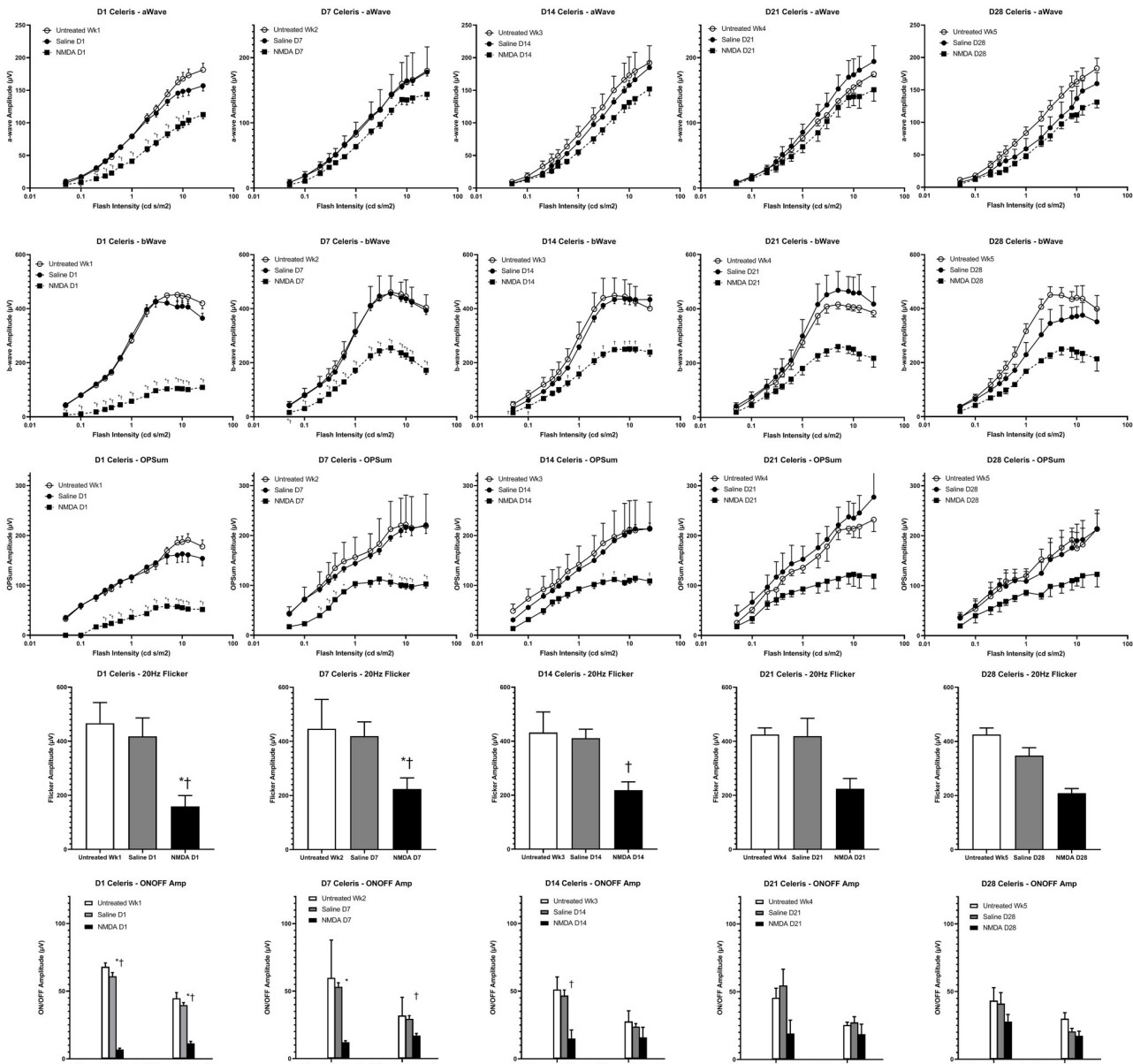

**Fig 5. ERG results of NMDA-damaged, vehicle-treated, and age-matched undamaged controls.** Undamaged chicks (n = 10 P7 to P21, n = 5 P28 to P35), NMDA-damaged chick eyes (OS), and saline vehicle-injected fellow chick eyes (OD) (n = 10, P7/D1 to P14/D7, n = 8, P21/D14, n = 6, P28/D21, n = 5, P35/D28) were examined with the Celeris ERG system. The data is organized by time (columns) and parameter (rows). From left to right, the columns represent one day post-injection (D1) and 7 days post-hatch (P7), D7/P14, D14/P21, D21/P28, and D28/P35. From top to bottom, the rows represent a-wave amplitudes, b-wave amplitudes, the amplitudes of the sum of oscillatory potentials (OPSum), flicker amplitudes at 20Hz, and ON/OFF bipolar cell response amplitudes. * indicates significance with vehicle (saline) controls, † indicates significance with undamaged controls.

sensitive than OFF bipolar cells to NMDA damage. However, the reason behind this difference will be investigated in future studies.

Data captured with the Celeris ERG system (Fig 5) was found to be similar to data captured with the UTAS ERG system (S3 Fig). This is notable as both systems use fundamentally different designs to capture the ERG waveforms. While the Celeris system uses dual electrode

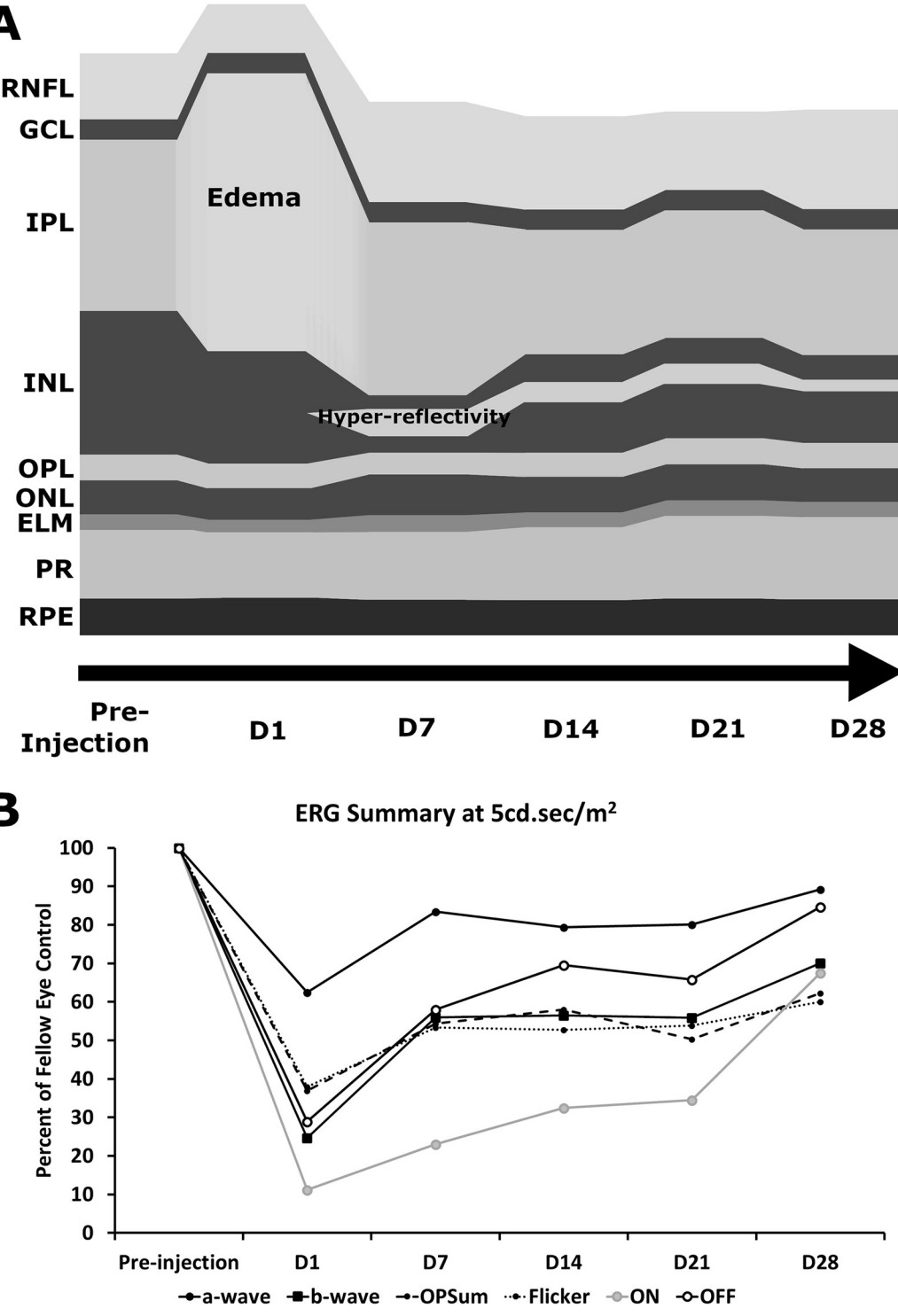

**Fig 6. Quantitative characteristic model of SD-OCT and ERG behavior during chick NMDA damage.** SD-OCT uses average layer thicknesses of NMDA-damaged chick eyes from D1 to D28 showing key features such as the edema present at D1 in the IPL and the hyperreflectivity present from D7 onwards in the INL (A). The ERG summary shows the average amplitudes as a percent of the fellow eye control at the 5 cd.sec/m² flash intensity from D1 to D28 (B).

stimulators that come into close contact with the ocular surface, the UTAS system uses the more traditional ganzfeld approach.

## Discussion

Despite the importance of the chick and the NMDA model for vision research, little normative data is available. This work is necessary to further develop the model to evaluate druggable targets for translational research. Herein we provide the first multimodal data on retinal structure and function in the chick NMDA model.

Interestingly we observed decreased ERG a-wave amplitudes at D1 despite a lack of TUNEL positive cells in the photoreceptor layers. Due to contributions of OFF-bipolar cells to the a-wave [33], this decrease in the a-wave amplitude could potentially be due to the disruption of the cone OFF-bipolar cells immediately after damage. While the ON-bipolar cells in our study were subject to the most lasting damage, the OFF-bipolar cells were only significantly reduced immediately after damage and recovered to near control amplitudes after D1. There is some contradictory data in the literature surrounding the effects of NMDA damage on the ON- and OFF- cone bipolar cells. One study of the effects of NMDA on rod and cone bipolar cells using a patch clamp technique found that NMDA significantly reduced response amplitudes of rod bipolar cells while the cone bipolar cells, both ON and OFF, were unaffected [34]. In contrast, another study observed almost complete ablation of both ON and OFF cone bipolar cells after combined NMDA and kainate damage [35]. Future studies may help resolve these discrepancies.

The most striking feature present in the SD-OCT imaging of the D1 NMDA retina was the retinal edema in the IPL. Following this initial swelling, the retina contracted considerably with the edema disappearing altogether, a behavior that has been noted in the established literature [25,36]. However, with this recovery and thinning of the IPL, the INL was observed to thicken slightly with the noted presence of a hyperreflective band on SD-OCT. As the edema on SD-OCT resolved at subsequent timepoints (Figs 4A and 6A), the a-wave amplitudes were noted to recover. Combining the edema resolution with the a-wave and OFF-bipolar cell recovery, we can hypothesize that the recovery of the edema is correlated with the a-wave and therefore OFF-bipolar cell recovery. Interestingly, the edema and hyperreflectivity in the inner retina with subsequent thinning is analogous to OCT scans of human retinal vascular occlusions [37–40].

In NMDA-damaged chick eyes, we observe significant levels of cell death in the INL consistent with the known loss of primarily amacrine cells in the inner retina [25,26,41,42]. Amacrine cells and the retinal ganglion cells (RGC) have been established as possessing NMDA-receptors, however in the chick model RGCs are rarely affected by NMDA damage [9,43–46]. While the other retinal neurons such as the photoreceptors, horizontal cells, and bipolar cells are generally considered to lack these receptors [9,47]. The loss of the amacrine cells, which are located in the INL and contribute to the oscillatory potentials [33], coincides with the large number of TUNEL positive cells in the INL and the amplitude reduction in the oscillatory potentials. In addition, we can see direct evidence of the retinal damage through the fundus imaging as the healthy pink color gives way to a retinal whitening, correlating with retinal edema and potentially further evidence of the ischemic conditions that lead to cell death. Despite lacking NMDA-receptors, bipolar cells are also strongly affected by NMDA damage through a potential TNFα glial cell pathway [48]. The b-wave amplitudes are not only severely diminished but the photopic hill effect is almost completely abrogated and remained muted even in subsequent timepoints. The photopic hill is a noted effect observed and heavily studied in human b-wave responses. It occurs due to the cone response of the bipolar cells with logistic

and gaussian curves [49–52]. Due to the diurnal nature of chicks, this effect can be directly observed (Fig 5—second row and Fig 6B) as the b-wave will reach a peak amplitude before starting to decline with further increases in flash intensity. This directly contrasts with the nocturnal models of mice and rats, which do not possess the cone density required for this effect.

While the chick model has many advantages for vision research, it does lack the breadth of genetic manipulation compared to more traditional mouse models. It is also unknown how repeated weekly anesthesia events could affect ocular parameters, particularly on the ERG. While efforts were taken to minimize diurnal changes, there remains the possibility that ERG results could shift in the hours between data acquisition. Additionally, due to electrode malfunctions, an additional batch of chicks had to be used to complete the D14/P21 and D21/P28 timepoints which could explain the discrepancies observed in ERG at those timepoints.

In conclusion, with this multimodal analysis of baseline chick ocular responses to NMDA excitotoxic damage, the stage is set for novel therapeutics to be tested. The chick eye proves to be more analogous to the human eye than other standard lab animal models between an area centralis, cone density, and the photopic hill effect. With the range of clinical disorders the chick NMDA model simulates including observable similarities to vascular occlusions, this could be a low-cost, simple model for high-throughput studies to develop and translate effective therapies to prevent vision loss.

## Supporting information

**S1 Fig. SD-OCT marker positions and fundus projection.** (A) Representative capture of chick retinal layers with the pecten shown in the center. (B) Layer identifiers used by the InVivoView Diver software to measure retinal thicknesses. (C) Representative fundus capture from volumetric OCT scan. Blue crosses represent where measurements were taken. D) Measurements of all retinal thicknesses of the RNFL, IPL, RNFL-IPL, INL, OPL, ONL, ELM, PR, RPE, and full retina layers between NMDA and saline (vehicle) treated eyes at P7/D1 to P35/D28 post injection and their respective age-matched untreated controls. Error bars are shown as standard deviation. Abbreviations: RNFL—retinal nerve fiber layer, GCL—ganglion cell layer, IPL—inner plexiform layer, INL—inner nuclear layer, OPL—outer plexiform layer, ONL—outer nuclear layer, ELM—external limiting membrane, PR—photoreceptors, IS—inner segment, OS—outer segment, ETPRS—end tip of photoreceptors, RPE—retinal pigment epithelium.
(TIF)

**S2 Fig. Weight measurements.** A normative database of undamaged and NMDA-damaged chicks were weighed weekly up until sacrifice at P35. The two treatment groups were typically weighed within one day of each other except for the P21/D14 timepoint where the NMDA-damaged chicks were measured three days after the undamaged chicks due to experimental constraints (n = 10/group).
(TIF)

**S3 Fig. In vivo ERG UTAS system measurements.** Undamaged chicks (n = 3), NMDA-damaged chick eyes (OS), and saline vehicle-injected fellow chick eyes (OD) (n = 3) were examined with the In Vivo ERG UTAS System. The data is organized by time (columns) and parameter (rows). From left to right, the columns represent one day post-injection (D1) and 3 weeks post-hatch (PWk3), D6/PWk4, and D13/PWk5. From top to bottom, the rows represent a-wave amplitudes, b-wave amplitudes, the amplitudes of the sum of oscillatory potentials (OPSum), flicker amplitudes at 20Hz, and ON/OFF bipolar cell response amplitudes. *

indicates significance with vehicle (saline) controls, † indicates significance with undamaged controls.
(TIF)

**S1 File.**
(ZIP)

**S2 File.**
(ZIP)

## Acknowledgments

The authors would like acknowledge the support of Dr. Andy J. Fischer, Isabella Palazzo, and Dr. Ralph Malbrue for their expertise with the chick model. The authors would like to thank Mahsa Adib, Haanya Ijaz, Yushin Jeng, and Mohamed Soumakieh for their technical assistance.

## Author Contributions

**Conceptualization:** Tyler Heisler-Taylor, Julie Racine, Colleen M. Cebulla.

**Data curation:** Tyler Heisler-Taylor, Elizabeth G. Urbanski, Colleen M. Cebulla.

**Formal analysis:** Tyler Heisler-Taylor, Richard Wan, Hailey Wilson.

**Investigation:** Tyler Heisler-Taylor, Richard Wan, Elizabeth G. Urbanski, Sumaya Hamadmad, Mohd Hussain Shah, Hailey Wilson.

**Methodology:** Tyler Heisler-Taylor, Richard Wan, Elizabeth G. Urbanski, Sumaya Hamadmad, Julie Racine.

**Project administration:** Colleen M. Cebulla.

**Resources:** Julie Racine, Colleen M. Cebulla.

**Software:** Tyler Heisler-Taylor, Julie Racine.

**Supervision:** Julie Racine, Colleen M. Cebulla.

**Validation:** Tyler Heisler-Taylor.

**Visualization:** Tyler Heisler-Taylor.

**Writing – original draft:** Tyler Heisler-Taylor.

**Writing – review & editing:** Tyler Heisler-Taylor, Richard Wan, Elizabeth G. Urbanski, Sumaya Hamadmad, Mohd Hussain Shah, Hailey Wilson, Julie Racine, Colleen M. Cebulla.

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
