## [Decision Letter · Decision Letter 0]

23 Jun 2021

PONE-D-21-15983

Multimodal imaging and functional analysis of the chick NMDA retinal damage model

PLOS ONE

Dear Dr. Cebulla,

Thank you for submitting your manuscript to PLOS ONE. After careful consideration, we feel that it has merit but does not fully meet PLOS ONE’s publication criteria as it currently stands. Therefore, we invite you to submit a revised version of the manuscript that addresses the points raised during the review process.

My recommendation is that all points raised by reviewers should de carefully addressed.

We look forward to receiving your revised manuscript.

Kind regards,

Alexandre Hiroaki Kihara, Ph.D.

Academic Editor

PLOS ONE

Journal Requirements:

1. Please ensure that your manuscript meets PLOS ONE's style requirements, including those for file naming. The PLOS ONE style templates can be found athttps://journals.plos.org/plosone/s/file?id=wjVg/PLOSOne_formatting_sample_main_body.pdf and https://journals.plos.org/plosone/s/file?id=ba62/PLOSOne_formatting_sample_title_authors_affiliations.pdf

Additional Editor Comments (if provided):

Reviewers' comments:

Reviewer's Responses to Questions

**Comments to the Author**

1. Is the manuscript technically sound, and do the data support the conclusions?

Reviewer #1: Yes

Reviewer #2: Yes

2. Has the statistical analysis been performed appropriately and rigorously? 

Reviewer #1: Yes

Reviewer #2: I Don't Know

3. Have the authors made all data underlying the findings in their manuscript fully available?

Reviewer #1: Yes

Reviewer #2: Yes

4. Is the manuscript presented in an intelligible fashion and written in standard English?

Reviewer #1: Yes

Reviewer #2: Yes

5. Review Comments to the Author

Reviewer #1: This manuscript showed functional and structural changes in NMDA-induced retinal damage in chick. The authors showed comprehensive data and they are convincing. Only minor points concerned are below:

Line 222. Only D1 timepoint showed a significant difference (p = 0.0005), but Figure 2 showed that D1 and D7 graphs are similar. Please check p value in D7. Also, please insert the asterisk in Figure 2.

Please indicate total animal number in the Methods section.

Supplemental figure has been changed by the authors. S2 included (D) as all layers of retinal thickness. Therefore, line 257, Other layers were found to be unchanged (Fig S3) should be (Fig S2D). Similarly, line 324, the UTAS ERG system (Fig S4) should be (Fig S3). Accordingly, supplemental figure legends should be corrected.

Line 371. As the edema on SD-OCT resolved at subsequent timepoints (Fig 4A) may be (Fig 6A).

Line 394 (Fig 6, second row) should be (Fig 6B).

Reviewer #2: 1. Suggested for authors to add update references in addition to references 2-6 to strengthen the fact that the indicated pathway is still valid.

2. Suggested for authors to include literature on in vitro retinal disease model, and why animal model is still necessary or preferred in such research.

3. It is generally discouraged in eye research to perform visually disabling treatments to both eyes. This issue is raised in the ARVO Statement “a visually disabling procedure should not be performed bilaterally unless the two procedures are related and unavoidable components of a specific project”. Please justify in an explicit manner, and aligned with current animal welfare policies.

4. Are there any chicks which later had to be excluded because of damage from, for instance, the IVT needle nicking the lens and causing a cataract, or develop infection? Please state.

5. What animal gender were used, and why?

6. Please make explicit the statistical methods employed to evaluate the differences. Do not assume by default normal distribution of datasets. Provide evidence for normality or use statistic tests that do not assume normality.

7. Line 216 - and clear presence OF the pecten ?

8. Line 220 - any reasons why n=10 for D1-D14, and only n=5 for D21-D28 ?

9. Line 268 & 269: effected or affected?

10. Figure 1 - What about representative image for vehicle-treated chicks?

11. Figure 4 - Why is there no representative image for undamaged chicks?

12. Please be consistent in using hyphen when writing connected words such as:

- vehicle-treated

- vehicle-injected

- NMDA-treated

- NMDA-damaged

Some phrases are with hyphen, some without

6. PLOS authors have the option to publish the peer review history of their article (what does this mean?). If published, this will include your full peer review and any attached files.

Reviewer #1: No

Reviewer #2: No

---

## [Author Response · Author response to Decision Letter 0]

6 Aug 2021

We thank the reviewers for their time and constructive comments. We have addressed each of their concerns below. 

Reviewer #1: This manuscript showed functional and structural changes in NMDA-induced retinal damage in chick. The authors showed comprehensive data and they are convincing. Only minor points concerned are below:

Line 222. Only D1 timepoint showed a significant difference (p = 0.0005), but Figure 2 showed that D1 and D7 graphs are similar. Please check p value in D7. Also, please insert the asterisk in Figure 2.

The p-value between Undamaged P14 and NMDA D7/P14 was 0.0988 according to the Tukey-Kramer HSD multiple comparison tests. We have included an asterisk on figure 2. The retesting of the data using non-parametric methods as requested by Reviewer 2 maintained the non-significance of the one week post damage condition with a p-value of 0.2690 and the significance of the D1 post damage condition at p=0.0075. These updated values have been included in the manuscript.

Please indicate total animal number in the Methods section.

These studies utilized a total of 38 chicks. This has been added to the methods section. 

“These studies utilized 4 chicks for TUNEL analysis, 20 chicks for the initial round of ERG/OCT/etc testing, and 14 chicks for the supplemental ERG/OCT/etc testing making a total of 38 chicks.”

Supplemental figure has been changed by the authors. S2 included (D) as all layers of retinal thickness. Therefore, line 257, Other layers were found to be unchanged (Fig S3) should be (Fig S2D). Similarly, line 324, the UTAS ERG system (Fig S4) should be (Fig S3). Accordingly, supplemental figure legends should be corrected.

Thank you for catching this. These errors have been corrected. The legend text for the previous S3 has been merged into the legend for S2 as part D. What had been labeled as S4 has been corrected to be the new S3. The merged S2 and S1 were swapped for position. It now reads:

“Fig S1: SD-OCT marker positions and fundus projection: (A) Representative capture of chick retinal layers with the pecten shown in the center. (B) Layer identifiers used by the InVivoView Diver software to measure retinal thicknesses. (C) Representative fundus capture from volumetric OCT scan. Blue crosses represent where measurements were taken. D) Measurements of all retinal thicknesses of the RNFL, IPL, RNFL-IPL, INL, OPL, ONL, ELM, PR, RPE, and full retina layers between NMDA and saline (vehicle) treated eyes at P7/D1 to P35/D28 post injection and their respective age-matched untreated controls. Error bars are shown as standard deviation. Abbreviations: RNFL - retinal nerve fiber layer, GCL – ganglion cell layer, IPL - inner plexiform layer, INL - inner nuclear layer, OPL - outer plexiform layer, ONL - outer nuclear layer, ELM – external limiting membrane, PR – photoreceptors, IS - inner segment, OS - outer segment, ETPRS - end tip of photoreceptors, RPE - retinal pigment epithelium.”

Line 371. As the edema on SD-OCT resolved at subsequent timepoints (Fig 4A) may be (Fig 6A).

We feel that 4A and 6A are both appropriate figure references as 4A shows the actual b-scans of the edema while 6A shows the representative model. The text has been changed to refer to both figures.

“As the edema on SD-OCT resolved at subsequent timepoints (Figs 4A and 6A), …”

Line 394 (Fig 6, second row) should be (Fig 6B).

It should have read Fig 5 – second row but 6B is also appropriate to display the b-wave amplitudes. The text has been corrected to refer to both figures.

“…this effect can be directly observed (Figs 5 - second row and 6B) as the b-wave… ”

Reviewer #2: 1. Suggested for authors to add update references in addition to references 2-6 to strengthen the fact that the indicated pathway is still valid.

We had originally intended to refer to those articles where the phenomenon was initially documented or at least close to it. We have updated our references to include additional and more recent publications.

“2. Campbell WA, Blum S, Reske A, Hoang T, Blackshaw S, Fischer AJ. Cannabinoid signaling promotes the de-differentiation and proliferation of Muller glia-derived progenitor cells. Glia. 2021.

3. Campbell WA, Fritsch-Kelleher A, Palazzo I, Hoang T, Blackshaw S, Fischer AJ. Midkine is neuroprotective and influences glial reactivity and the formation of Muller glia-derived progenitor cells in chick and mouse retinas. Glia. 2021;69(6):1515-39.

9. Shen Y, Liu XL, Yang XL. N-methyl-D-aspartate receptors in the retina. Mol Neurobiol. 2006;34(3):163-79.

10. Dong XX, Wang Y, Qin ZH. Molecular mechanisms of excitotoxicity and their relevance to pathogenesis of neurodegenerative diseases. Acta Pharmacol Sin. 2009;30(4):379-87.

11. Kuehn S, Rodust C, Stute G, Grotegut P, Meiner W, Reinehr S, et al. Concentration-Dependent Inner Retina Layer Damage and Optic Nerve Degeneration in a NMDA Model. J Mol Neurosci. 2017;63(3-4):283-99.

12. Connaughton V. Glutamate and Glutamate Receptors in the Vertebrate Retina. In: Kolb H, Fernandez E, Nelson R, editors. Webvision: The Organization of the Retina and Visual System. Salt Lake City (UT)1995.”

2. Suggested for authors to include literature on in vitro retinal disease model, and why animal model is still necessary or preferred in such research.

While in vitro models of retinal disease can be useful when employed correctly, the subject of this manuscript deals solely with in vivo animal models. Primarily with the chick model but the introduction specifically compares with the more traditional mouse model. An in vitro model cannot accurately simulate the complex structure and interactions occurring within the retina that could affect how NMDA models glutamate excitotoxicity. 

3. It is generally discouraged in eye research to perform visually disabling treatments to both eyes. This issue is raised in the ARVO Statement “a visually disabling procedure should not be performed bilaterally unless the two procedures are related and unavoidable components of a specific project”. Please justify in an explicit manner, and aligned with current animal welfare policies.

While bilateral NMDA damage is unfortunately necessary in some cases to provide a contralateral control. We have unpublished work showed variability of NMDA unilaterally injected chicks when we evaluated experimental agents; it was determined that bilateral treatment groups would provide an internal control. However, during this work, no animals were subjected to bilateral NMDA damage. The left eye was given the NMDA damage while the right eye was given the sterile saline vehicle. Sterile saline is one of two approved vehicle solutions, the other being sterile water, for clinical human injections due to the lack of adverse effects on the eye. Observations taken of eyes injected with sterile saline only show an acute immune reaction due to the physical injection of the needle. Our data show how the sterile saline treated eyes were indistinguishable from completely undamaged chicks. Furthermore, the chicks are monitored daily by ULAR veterinarians for infection or other adverse effects. Any chicks that meet the early removal criteria are promptly euthanized by either lab personnel or ULAR veterinarians. During experimental procedures and measurements, the chick eyes are inspected for infections or hemorrhage. Additionally, we have also shown no changes in feeding or weight gain over time compared to untreated or vehicle controls. 

4. Are there any chicks which later had to be excluded because of damage from, for instance, the IVT needle nicking the lens and causing a cataract, or develop infection? Please state.

These are both within our early removal criteria. Fortunately chicks are not as susceptible to lens nicking due to the comparative size of the eye to lens and none in this study experienced lens damage or infection. Additionally, no hemorrhage was noted in the eyes. During each measurement time point, chicks were examined with a slit lamp for any damage to the anterior chamber and with an operating microscope to check for retina hemorrhage or other damage. 

5. What animal gender were used, and why?

Male chicks were used as they were the most readily available from our supplier. Female chicks are more desired for their ability to produce marketable eggs. 

6. Please make explicit the statistical methods employed to evaluate the differences. Do not assume by default normal distribution of datasets. Provide evidence for normality or use statistic tests that do not assume normality.

Statistics have been changed to utilize non-parametric tests. Specifically, Kruskal-Wallis test was used to evaluate differences between experimental groups and the Steel-Dwass method was utilized in JMP to perform non-parametric multiple comparisons. Graphs and statistics mentioned in the text have been updated to reflect this change. 

7. Line 216 - and clear presence OF the pecten ?

Thank you for finding this grammatical mistake, it has been corrected.

8. Line 220 - any reasons why n=10 for D1-D14, and only n=5 for D21-D28 ?

At the mid-point of the experiment (D14), half of the subjects were sacrificed to provide tissue for frozen sections. This was done so that we would have frozen sections of D1 for TUNEL, D14, and D28. 

9. Line 268 & 269: effected or affected?

Thank you for pointing out this grammatic error, it has been changed to ‘affected’. It’s amazing what you miss even after reading through it a dozen times. 

10. Figure 1 - What about representative image for vehicle-treated chicks?

Vehicle and undamaged chicks were indistinguishable upon inspection. They were combined into the category of ‘healthy’ or ‘normal’ chicks. 

11. Figure 4 - Why is there no representative image for undamaged chicks?

Similarly to the previous question, the vehicle and saline treated eyes were virtually indistinguishable as can be seen from the numerical data in part D of the figure. We chose to leave out the undamaged b-scan so that the vehicle and NMDA treated eyes could be shown larger. 

12. Please be consistent in using hyphen when writing connected words such as:

- vehicle-treated

- vehicle-injected

- NMDA-treated

- NMDA-damaged

Some phrases are with hyphen, some without

All non-hyphenated instances of the above phrases have been hyphenated.

---

## [Decision Letter · Decision Letter 1]

25 Aug 2021

Multimodal imaging and functional analysis of the chick NMDA retinal damage model

PONE-D-21-15983R1

Dear Dr. Cebulla,

We’re pleased to inform you that your manuscript has been judged scientifically suitable for publication and will be formally accepted for publication once it meets all outstanding technical requirements.

Kind regards,

Alexandre Hiroaki Kihara, Ph.D.

Academic Editor

PLOS ONE

Additional Editor Comments (optional):

Reviewers' comments:

Reviewer's Responses to Questions

**Comments to the Author**

1. If the authors have adequately addressed your comments raised in a previous round of review and you feel that this manuscript is now acceptable for publication, you may indicate that here to bypass the “Comments to the Author” section, enter your conflict of interest statement in the “Confidential to Editor” section, and submit your "Accept" recommendation.

Reviewer #1: (No Response)

Reviewer #2: All comments have been addressed

2. Is the manuscript technically sound, and do the data support the conclusions?

Reviewer #1: (No Response)

Reviewer #2: Yes

3. Has the statistical analysis been performed appropriately and rigorously? 

Reviewer #1: (No Response)

Reviewer #2: Yes

4. Have the authors made all data underlying the findings in their manuscript fully available?

Reviewer #1: (No Response)

Reviewer #2: Yes

5. Is the manuscript presented in an intelligible fashion and written in standard English?

Reviewer #1: (No Response)

Reviewer #2: Yes

6. Review Comments to the Author

Reviewer #1: (No Response)

Reviewer #2: (No Response)

7. PLOS authors have the option to publish the peer review history of their article (what does this mean?). If published, this will include your full peer review and any attached files.

Reviewer #1: No

Reviewer #2: **Yes: **Izuddin Fahmy Abu

---

## [Editor Report · Acceptance letter]

27 Aug 2021

PONE-D-21-15983R1 

Multimodal imaging and functional analysis of the chick NMDA retinal damage model 

Dear Dr. Cebulla:

I'm pleased to inform you that your manuscript has been deemed suitable for publication in PLOS ONE. Congratulations! Your manuscript is now with our production department. 

Kind regards, 

on behalf of

Dr. Alexandre Hiroaki Kihara 

Academic Editor

PLOS ONE